# Health Status of *Mytilus chilensis* from Intensive Culture Areas in Chile Assessed by Molecular, Microbiological, and Histological Analyses

**DOI:** 10.3390/pathogens11050494

**Published:** 2022-04-21

**Authors:** Pablo Santibáñez, Jesús Romalde, Derie Fuentes, Antonio Figueras, Jaime Figueroa

**Affiliations:** 1Programa de Doctorado en Ciencias de la Acuicultura, Facultad de Ciencias, Universidad Austral de Chile, Los Pinos s/n, Balneario Pelluco, Puerto Montt 5110566, Chile; 2Interdisciplinary Center for Aquaculture Research (INCAR), Concepción, Bío-Bío 4030000, Chile; jefigueroa@uach.cl; 3Department of Microbiology and Parasitology, CRETUS & CIBUS-Faculty of Biology, Universidade de Santiago de Compostela, 15782 Santiago de Compostela, Spain; jesus.romalde@usc.es; 4Bio-Computing and Applied Genetics Division, Center for Systems Biotechnology, Fraunhofer Chile Research Foundation, Santiago 8580704, Chile; Derie.Fuentes@fraunhofer.cl; 5Institute of Marine Research (IIM), National Research Council (CSIC), Eduardo Cabello 6, 36208 Vigo, Spain; antoniofigueras@csic.es; 6Department of Biochemistry and Microbiology, Faculty of Biochemistry, University Austral of Chile, Valdivia, Los Ríos 5091000, Chile

**Keywords:** mussel, aquaculture, pathogens, eukaryotic communities, 18S rDNA

## Abstract

Shellfish farming is a relevant economic activity in Chile, where the inner sea in Chiloé island concentrates 99% of the production of the mussel *Mytilus chilensis*. This area is characterized by the presence of numerous human activities, which could harm the quality of seawater. Additionally, the presence of potentially pathogenic microorganisms can influence the health status of mussels, which must be constantly monitored. To have a clear viewpoint of the health status of *M. chilensis* and to study its potential as a host species for exotic diseases, microbiological, molecular, and histological analyses were performed. This study was carried out in October 2018, where *M. chilensis* gut were studied for: presence of food-borne bacteria (*Vibrio parahaemolyticus*, *Escherichia coli*, *Salmonella* spp.), exotic bacteria (“*Candidatus Xenohaliotis californiensis*”), viruses (abalone and Ostreid herpes virus), and protozoa (*Marteilia* spp., *Perkinsus* spp. and *Bonamia* spp.). Additionally, 18S rDNA metabarcoding and histology analyses were included to have a complete evaluation of the health status of *M. chilensis.* Overall, despite the presence of risk factors, abnormal mortality rates were not reported during the monitoring period and the histological examination did not reveal significant lesions. Pathogens of mandatory notification to World Organization for Animal Health (OIE) and the Chilean National Fisheries and Aquaculture Service (SERNAPESCA) were not detected, which confirms that *M. chilensis* have a good health status, highlighting the importance of an integrated vision of different disciplines to ensure the sustainability of this important mussel industry in Chile.

## 1. Introduction

Increasing social apprehension about the environmental quality and vulnerability of biodiversity of the marine coastal areas have been observed in recent years, both on a global and local scale [1,2]. Mussels of the genus *Mytilus*, are widely distributed throughout the oceans and are typically found in cold waters in both hemispheres, following an antitropical distribution pattern [3]. They are important components of coastal ecosystems [4] and objects of research as cosmopolitan inhabitants of high-latitude coastal marine ecosystems. In addition to their ecological significance, they are commercially important in the global scenario with the Chilean mussel (*M. chilensis*) ranking in the top five mollusk species intended for aquaculture production worldwide [5].

*Mytilus chilensis* (Hupé 1854) belongs to the group of marine mollusks included in the Bivalvia class. It recently acquired the definitive status of a species recognized and accepted in the World Register of Marine Species (WoRMS) with the AphiaID: 397,041 [6]. It is distributed along approximately 1900 km of the Chilean coast from the Gulf of Arauco (35° S) in the north to Cape Horn (55° S) in the south, inhabiting intertidal and subtidal environments down to 10 m in depth [7]. Currently, 99% of the total commercial production of *M. chilensis* comes from suspended cultures in large-scale mussel farms located mainly between 41°30″ S and 43°30″ S, within the protected Chiloé inner sea, in southern Chile. The Chilean mussel industry has grown exponentially in the last 20 years, with current production at 338,000 tons/year, which makes *M. chilensis* the second most important product of Chilean aquaculture, after Atlantic salmon [8].

Harvesting of *Mytilus* spp. mussels throughout the world is based on the exploitation of both wild and cultivated populations [9]. Mussel farming is based on long-line systems suspended either at the surface or sub-surface, in coastal or open waters. The Chilean mussel farms are in highly anthropized areas in the inner sea of Chiloé, characterized by intensive aquaculture and tourist activities. Although these guarantee a good supply of nutrients for mussels breeding, they could represent a problem due to the introduction of contaminants that can weaken the immune system of the mussels. The presence of biotic and abiotic stressors agents can lead to a critical situation, depending on the seasonality or the performance of human activities at specific sites. Although no mass mortality events associated to pathogens have been reported in Chile, events have been reported in mussels in other parts of the world [10].

To continue previous studies [11] and contribute to the definition of the health status of the Chilean mussel, we report the results of a multidisciplinary study, evaluating pathogens that can affect the welfare of marine species with economic importance, as well as public health, using *M. chilensis* as a sentinel species. The target microorganisms are defined in the list of exotic diseases of the World Organisation for Animal Health (OIE) and Chilean National Fisheries and Aquaculture Service (SERNAPESCA), supplemented with 18S rDNA metabarcoding, histology, and foodborne pathogens analyses. This study provides a new and comprehensive perspective on the health status of *M. chilensis* to provide baseline data for future reference in the mussel aquaculture regions of Patagonia.

## 2. Results

### 2.1. Microbiological Analysis

A total of three mussel farms (S1–S3) were sampled for microbiological, histological, and molecular analysis (Figure 1, Table 1). Analysis of foodborne pathogens in this study corroborated previous findings from our laboratory, where *Salmonella* spp. and *V. parahaemolyticus* were not detected in mussel farms. The *E. coli* bacterium was detected in all the samples with the most probable number analysis (MPN), all with contamination levels <100 MNP/100 g (Table 2). This finding indicates that the mussels are of a good quality for harvesting, all mussel farms being classified as category A.

### 2.2. Pathogens Evaluation and Histological Analyses

No mass mortality events were observed during this study. Samples from sites S1–S3, are in good external condition. All samples were PCR negative for abalone (AbHV) and Ostreid herpesvirus 1 (OsHV-1), parasites (*Bonamia* spp., *Marteilia refringens* and *Perkinsus* spp.) and bacteria (*Candidatus Xenohaliotis californiensis*) (Table 3). Histological analysis described that the common finding in sites S1-S3 was the presence of intracellular bacteria in the digestive tract in low to moderate intensity (prevalence S1: 20%, S2: 36.7% and S3: 33.3%). Additionally, it was observed that intracellular bacteria did not cause alteration in the digestive tract that could affect its correct functioning, observing no structural anomaly (data not shown). These histological results were consistent with the field observations and the PCR results described above.

### 2.3. General Sequencing Results

All samples obtained from gut tissue (n = 10) failed the PCR amplification [12,13,14]. From the gut content samples (n = 10), DNA was successfully obtained in five of them, three from mussels collected in site 3 (MF group) and two from natural habitat from site 4 (Wild Type group, WT). A total of 5,017,389 raw data reads were generated by Illumina MiSeq sequencing. 2,818,830 high-quality reads remained after trimming and filtering, with a range between 506,051 and 629,065 of non-chimeric reads per sample (Table 4). The rarefaction curves show the number of species as a function of the number of reads, initially growing rapidly as the most common species are found and then stabilizing as only the less frequent species remain. All rarefaction curves of observed species richness reached saturation (Figure 2). Differences in sequencing efficiency were observed in the samples analyzed. The classification against SILVA reference database assigned the reads to 79 families and revealed the presence of 116 genera, where approximately 20% of the sequences could not be assigned to any eukaryote genus. A Tukey’s HSD (Honestly-significant-difference) test showed that the WT group were significantly different from the MF group (*p* < 0.05).

### 2.4. Comparison between 18S rDNA Gut Eukaryotic Communities from WT and MF Groups of M. chilensis 

A high proportion of host DNA sequences were detected in the samples, highlighting the importance of having specific nucleic acid extraction protocols to reduce contamination with the host species. The analysis of the eukaryotic communities showed that the Simpson index was close to 1 in all samples analyzed, showing high community diversity (Table 4). Figure 3 shows the relative abundance of the 10 most-represented eukaryotic organisms at the genus level found in the mussels gut. All samples have a significant abundance of 18S rDNA sequences from host tissue, ranging from 68.8% to 79.02% of the total sequences. The number of unidentified 18S rDNA reads, representing new species, remained high for all samples tested, close to 20%. The relative abundance of sequences that have been assigned to eukaryotic organisms ranges from 0.2% to 13.95% (Appendix A). 

In general, the eukaryotic composition in both groups was dominated by few genera, which represents the majority of reads in the analyzed samples. Dominant genera were different between the WT and MF groups, showing differences in their relative frequencies as shown in a heat-map (Figure 4). From the sequences that could be assigned, the Peridiniales family have a greater abundance than the other families of the WT group, not being detected in individuals from the MF group. Despite the low abundances obtained in most from the eukaryotic organisms detected, this study was able to determine the presence of 116 different genera that are part of the gut eukaryotic community of *M. chilensis*. This is influenced by the site-dependent feeding of the samples and provides preliminary information on the possible composition of the surrounding plankton. Further studies are needed to establish the relationship of plankton with the eukaryotic composition of the gut of mussels.

The NMDS plot showed that the MF and WT groups could be differentiated by their gut 18S rDNA eukaryotic composition (Figure 5). The distinct eukaryotic communities were evident from the beta analyses, showing that the composition was similar within each group, but different between them. In general, bivalve samples from mussel farms showed lower diversity and richness of the associated eukaryotic communities compared to samples from natural habitats. Although the two groups share many species, the WT group shows more different genera than the MF group (Figure 6), having 34 genera exclusive to this group, the genus *Heterocapsa* being the most important. 

## 3. Discussion

Regarding histological analysis, intracellular prokaryotic inclusions were broadly distributed in the analyzed mussels, with the highest prevalence in Site S2 (36,7%). This finding was not related to any alteration in the digestive epithelium affecting its normal functioning. Additionally, no mass mortalities were recorded in the period of this study. The occurrence of prokaryotic inclusions in the epithelial cells of the gills and digestive gland tubules is widespread among mollusks [15,16,17]. Some histopathologic effects caused by prokaryotic inclusions have been reported previously in bivalves [15]. In *M. chilensis* from southern Chile, the multiplication of the bacteria caused hypertrophy of the infected gill epithelium cells, although low values of prevalence and intensity of infection were recorded [18].

The comparative analyses of three geographical sites characterized by the presence the mussel farms (Sites 1–3) showed that *M. chilensis* had an adequate health status, without the presence of exotic pathogens, significant histological findings, or critical counts of microorganisms dangerous to public health. Protozoa parasites of the genera *Perkinsus*, *Marteilia* and *Bonamia* were recognized as the main challenges for populations of natural and cultured bivalves [19]. However, since their identification, the publications and studies have not reflected the high environmental and economic impact that they have [20]. These species can infect abalones, oysters, clams, and mussels. For this reason, they are currently under mandatory notification to the OIE [21]. Since the first description of *Marteilia* (Paramyxea) in the flat oyster *Ostrea edulis* in 1968 in the Aber Wrach, Brittany (France), the life cycle of this parasite has remained unknown. *M. refringens* is one of the most significant pathogens of bivalve mollusks, with two species “O” and “M” based on polymorphisms in the internal transcribed spacer region of the ribosomal RNA genes [22]. *Marteilia* was detected with low prevalence in *M. galloprovincialis* and *M. edulis* [23,24,25], having a significant negative effect on the growth rate and length in *M. galloprovincialis* [26]. On the other hand, during a study on the mussel *M. galloprovincialis* in Tokyo Bay, infection by the protozoan parasite *Perkinsus beihaiensis* and *P. olseni* was found by histological examination and PCR analyses [27]. However, no mass mortality on account of *Perkinsus* has been described in mussels, where *Mytilus* plasma probably has a protecting role [28]. Finally, *Bonamia* sp. has not been detected in species of the genus *Mytilus* [29]; however, in Chile, it was diagnosed in the *Ostrea chilensis* oyster [30,31]. The possible relationship with other putative intermediate hosts such as *M. chilensis* is relevant. In this study, presence of *M. refringens*, *Perkinsus* spp., and *Bonamia* spp. parasites was not detected in the samples analyzed by PCR. Furthermore, no histopathological abnormalities were detected in the individuals studied, and 18S rDNA reads were not taxonomically assigned to these parasitic genera. Interestingly, 18S rDNA metabarcoding identified the genus *Parvilucifera* [32] of the family Perkinsidae in one sample from WT group (Appendix A), *Parvilucifera* being an alveolate that parasite dinoflagellates [33].

The AbHV and OsHV-1 are important pathogens in abalones and oysters, respectively [34,35]. The genus *Mytilus* can harbor the OsHV-1, being considered a reservoir or host for this virus without histological abnormalities [36,37,38,39,40]. In this study, the OsHV-1 and AbHV viruses were not detected in the samples analyzed by PCR. Furthermore, no histopathological abnormalities were detected in the individuals studied. To date, to our knowledge there are no reports of AbHV and OsHV-1 in *M. chilensis*, in accord with our results. 

Regarding bacterial analysis, withering syndrome is a fatal disease in abalones, caused by a Rickettsiales-like “*Candidatus Xenohaliotis californiensis*”. However, as infected abalones have been transported to Chile and others countries [41], *X. californiensis* was surveyed in this study. DNA of this bacterium was not detected in the individuals analyzed. However, a previous study using 16S rDNA metabarcoding found sequences from *Xenohaliotis* in the gut of mussels collected from mussel farms in Chile [11]. Futher studies should be carried out to clarify the relationship between *M. chilensis* and *Xenohaliotis*.

High-throughput sequencing enables microbial community structure to be analyzed with higher taxonomic resolution. The most popular method for high-throughput sequencing is PCR amplicon sequencing of genetic markers, such as 18S rRNA genes for eukaryotes [42,43]. This approach contributes to our general understanding of environmental eukaryotic diversity and distribution [44]. In this study, 20% of the sequences could not be taxonomically assigned to any family, which reveals a great research opportunity to discover new eukaryotic species in the waters of southern Chile. In general, the eukaryotic composition in mussels was dominated by few genera, showing differences in their relative frequencies according to the origin of the individuals. The family Peridiniales have a greater abundance than the other families in the WT group, not being detected in individuals from the MF group (Site 3). This family of dinoflagellate organisms are numerous in plankton, where the genus *Heterocapsa* detected in this work, was previously associated with algal blooms [45,46]. In this work, for the first time to the best of our knowledge, the gut eukaryotic composition of the bivalve species *M. chilensis* was studied. Microbial communities associated with gut samples were significantly different depending on their location, thus indicating the existence of a host-specific interactions with the environment. In general, bivalve samples from mussel farm showed lower eukaryotic diversity when compared with samples from the natural habitats. We hypothesize that high densities of mussels in culture and proximity to other aquatic industries that provide significant amounts of nutrients to marine ecosystems and anthropogenic contamination, generate conditions for specific microorganism growth, reducing competition and favoring the selection of certain genera, which is reflected in the eukaryotic composition of *M. chilensis* gut.

Finally, to complement the health information of *M. chilensis*, a limited study of critical microbiological parameters that define the food quality of the mussel at harvest time was carried out. The presence of *Salmonella* spp. and *Vibrio parahaemolyticus*, both pathogens related to food safety, were negative in all samples analyzed. In addition, the determination of *Escherichia coli* by the most probable number method, showed that all samples have counts <100 MPN/100 g, which defines the extracting areas as class A category, according to European regulations being safe for extraction for commercial purposes. 

In Chile, the knowledge of mollusk diseases has few antecedents, being restricted to the description of the presence of some multicellular parasites, the report of cell proliferative disorder in Chiloé oysters [47], presence of *Bonamia* sp. in *Ostrea chilensis* [30], description of bacterial agents in larvae of the northern oyster and more recently in adults of *M. chilensis*, and intracellular bacteria-like organisms in the digestive gland and gills [17]. All analyses carried out in this study, seen together, allow us to conclude that the Chilean mussel, *M. chilensis*, has a good health status and quality for its commercialization. We have concluded that *Mytilus chilensis* is devoid of serious parasites or diseases. No pathogens were found in this study based on the lists of notifiable disease in the World Organization for Animal Health and National Fisheries and Aquaculture Service of Chile, no significant histological findings were detected, and the 18S rDNA metabarcoding analyzes did not identify possible pathogens of interest to the industry. Overall, this study provides a new and comprehensive perspective on the health status of *M. chilensis*, to provide baseline data for future investigations into mussel aquaculture.

## 4. Materials and Methods

### 4.1. Site Information and Sample Collection

In October 2018, approximately 150 adults mussels (*M. chilensis*, ≥5 cm in shell length) per site were collected from three mussel farms located in the inner sea of Chiloé Island (Sites 1–3) and from a natural habitat (site 4) located far from the mussel farms (Figure 1). The Chiloé area was chosen because it concentrates 99% of Chilean mussel production. Site S1 is located near the city of Puerto Montt, the capital of the Lake District, with a population of over 200,000 in habitants, characterized by intense anthropogenic activity. Site S2 is close to the Chacao channel, characterized by intense maritime traffic and, Site S3 is in the inner sea of Chiloe, characterized by intense aquaculture activity, with salmon and mussel farms coexisting between them. Finally, Site 4 was chosen due to low anthropogenic activities in the area, being considered a WT population for metagenomics analysis. 

Mussels were collected from a long line at 1 m of profundity, kept cool, and transported live to the laboratory for histological processing and DNA extraction. The microbiological and histological analyses were carried out in the laboratory in less than 24 h post sampling. The samples obtained from sites 1–3 were processed for microbiological, histological, and molecular (PCR) analysis. Additionally, samples from sites 3 and 4 were processed for 18S rDNA metabarcoding analysis. The GPS coordinates for each mussel farm, abiotic factors, such as seawater temperature, pH, oxygen dissolved levels and salinity were recorded at all sampling sites (Table 1). 

### 4.2. Microbiological Analysis

A total of five samples per site (S1–S3) were processing for microbiological analysis. Analysis for *Escherichia coli* ß-glucuronidase—positive determination was performed using the methodology based on ISO/TS 16649-3:2015: “Microbiology of food and animal feeding stuffs—Horizontal method for the enumeration of ß-glucuronidase—positive *Escherichia coli*” [48]. Analysis for *Vibrio parahaemolyticus* was performed using the FDA/bacteriological analytical manual, Chapter 9, Online, 2004 [49]. Finally, analysis for *Salmonella* spp. was performed using the ISO 6579:2017: “Microbiology of food and animal feeding stuffs—Horizontal method for the detection of *Salmonella* spp.” [50].

### 4.3. Histological Analysis

A total of 30 mussels per site (S1-S3) were processed for histological evaluations. The individuals were cleaned externally, and their valves were separated using a knife. The whole body of each mussel was removed and fixed for 48 h in Carson’s solution at room temperature. Subsequently, the samples were reduced, and cross sections containing gills, digestive gland, gonads, foot, mantle, and kidney were obtained. The fixed tissues were then dehydrated by ascending ethanol series and embedded in paraffin wax. Samples were cut in 4 ± 2 μm serial sections on rotary microtome and stained with standard haematoxylin-eosin (HE) staining to evaluate the histopathological features. The slides were evaluated microscopically with increasing magnifications (10×, 20× and 40×) by a light microscope Leica model DMLS (Leica, Wetzlar, Germany). The prevalence of intracellular bacteria in the digestive tract was recorded in all samples analyzed. 

### 4.4. Nucleic Acids Extraction

For the detection of viruses, bacteria, eukaryotic parasites and, 18S rDNA metabarcoding analysis, the epithelia of the digestive gland and gut content from mussels were sampled and deposited in 2 mL tubes containing 1 mL of lysis buffer and zirconia beads (0.1 mm). Each sample consisted of pooled tissues obtained from three mussels to complete 20 mg approximately. The tissue was chosen due to the probability of detecting the DNA target due to the high water filtration capacity of mussels -about 33–50 mL min^−1^, which allows for concentrating microorganisms surrounding many folds [51,52]. The pooled samples´ homogenization was performed using a beadbeater for 30 s at maximum speed and the DNA extraction was carried out using a commercial kit E.Z.N.A.^®^ Total DNA/RNA isolation Kit (Omega Bio-tek, Norcross, GA, USA) following the manufacturer’s instructions. Each sample of DNA extractions were stored at −20 °C until use in PCR reactions. All PCR reactions were performed in duplicate and negative and positive controls were included in all experiments. The primers sequences and PCR reaction conditions are summarized in Appendix A. The Me15/Me16 primers [53] were used to validate the nucleic acid extraction procedure before the pathogens screening by PCR.

### 4.5. Pathogen Analysis

#### 4.5.1. Virus Detection

The oyster and abalone farms are located close to mussel farms in Chile, for this reason, the presence of AbHV and OsHV-1 was investigated in *M. chilensis* as a possible reservoir. The AbHV DNA search was carried out according to World Organization for Animal Health using the primers ORF77F1/ORF77R1 and ORF77 FAM-probe [54], by a real time qPCR. Reactions contained 12.5 μL TaqMan^®^ Fast Universal PCR Master Mix 2x (Agilent, Santa Clara, CA, USA), 2 μL (~100 ng μL^−1^) of extracted DNA sample, and the reaction mix was made up to 25 μL with deionized water after the primers and probe being added. The following thermal cycling conditions were used: 95 °C for 10 min followed by 45 cycles of 95 °C for 3 s and 60 °C for 30 s, using an AriaMx thermocycler (Agilent, Santa Clara, CA, USA). The OsHV-1 DNA detection was carried out by an end point PCR using two primers pairs combinations for the DNA detection, the external C5/C13 combination which generated a 765 bp product, followed by the internal C2/C4 pair which yielded a 352 bp product [54,55]. The amplification reaction was performed in 25 μL containing 12.5 μL of TaqMan^®^ Fast Universal PCR Master Mix 2x, 2 μL (~100 ng μL^−1^) of extracted DNA sample, 100 μM of each primer and water to complete the final volume. After heating samples for 10 min at 95 °C, 35 cycles consisting of 95 °C for 1 min, 50 °C for 1 min and 72 °C for 1 min were carried out, followed by a final elongation step of 10 min at 72 °C, using a SimpliAmp thermocycler (Applied Biosystems, Waltham, MA, USA). If amplifications were observed, a nested PCR was carried out using 0.5 µL of the primary reaction as a template following identical reaction conditions. The PCR products were separated by electrophoresis on 2% agarose gel stained with Gel Green (Biotium, Hayward, CA, USA) and visualized using ultraviolet (UV) illumination.

#### 4.5.2. Bacteria Detection

Previous results from our laboratory identified the presence of *Xenohaliotis* sequences in the *M. chilensis* gut microbiome using 16S rDNA metabarcoding [11], therefore, the evaluation of “*Candidatus Xenohaliotis californiensis*” in this study is of interest. The PCR reactions were performed as is described in the manual, Manual of Diagnostic Test [54], using the primers R5-1/RA3.6. PCR amplification was performed in 25 µL reaction volume containing 12.5 μL TaqMan^®^ Fast Universal PCR Master Mix 2x, 2 μL (~100 ng μL^−1^), 0.5 µM of each primer, and water to complete the final volume. The reaction mixtures were cycled in a thermal cycler SimpliAmp (Applied Biosystems, Waltham, MA, USA) with an initial denaturation at 95 °C for 10 min, 40 cycles at 95 °C for 1 min, 62 °C for 30 s, and 72 °C for 30 s, and a final extension at 72 °C for 10 min. An aliquot of each PCR reaction product was checked by 2% agarose gel electrophoresis.

#### 4.5.3. Parasites Detection

Protozoa of the genus *Perkinsus* spp., *Marteilia* spp. and *Bonamia* spp. are recognized as the main challenges for natural and cultivated bivalve populations [19], being objects of study. The PCR analyses for *P. marinus* and *P. olseni* were performed using PerkITS85 and PerkITS750 primers that amplify the ITS fragment [56] as is described in the manual, Manual of Diagnostic Test for Aquatic Animals [54]. Each PCR reaction contained: 12.5 μL of TaqMan^®^ Fast Universal PCR Master Mix 2x, 2 μL each primer at 0.1 µM, 2 μL (~100 ng μL^−1^) DNA sample and water to complete 25 μL. Amplification conditions consisted of an initial denaturation at 95 °C for 10 min, followed by 40 cycles of 95 °C for 1 min, 55 °C for 1 min, 72 °C for 1 min, with a final elongation at 72 °C for 10 min. The PCR products were subsequently checked by 2% agarose gel electrophoresis.

The presence of *M. refringens* was evaluated using the published ITS1 primers Pr4-Pr5 [22] as is described in the manual, Manual of Diagnostic Tests for Aquatic Animals [54]. The PCR reactions contained 12.5 μL TaqMan^®^ Fast Universal PCR Master Mix 2x, 2 μL (~100 ng μL^−1^) DNA sample, 1 µM forward and reverse primers and water to complete 25 μL. The PCR program consisted of the initial denaturation of DNA at 95 °C for 10 min, 30 cycles are performed as follows: denaturation at 95 °C for 1 min, annealing at 55 °C for 1 min, and elongation at 72 °C for 1 min. A final elongation step of 10 min at 72 °C was performed. The PCR products were subsequently checked by 2% agarose gels electrophoresis.

For *Bonamia* spp. evaluation, PCR was performed using the BO/BOAS primers [57], as is described in the manual, Manual of Diagnostic Tests for Aquatic Animals [54]. The PCR mixtures contained 12.5 μL TaqMan^®^ Fast Universal PCR Master Mix 2x, 2 μL DNA sample (~100 ng μL^−1^), 1 µM forward and reverse primers, and water in a total volume of 25 µL. Samples were denatured in an AriaMx thermocycler for 10 min at 95 °C, followed by 30 cycles (95 °C for 1 min, 55 °C for 1 min, 72 °C for 1 min) with a final extension of 10 min at 72 °C. After amplification, PCR products were subsequently checked by 2% agarose gel electrophoresis.

### 4.6. 18S rDNA Metabarcoding

Mussels collected from sites 3 and 4 were chosen for 18S rDNA metabarcoding analysis due to the opposite characteristics of their habitats of origin. Individuals analyzed from site 3 were designated as MF (Mussel Farm) and from site 4 as WT. Nucleic acids were extracted from gut samples using PureLink microbiome DNA Purification kit (Invitrogen, Waltham, MA, USA) according to the manufacturer´s instructions. DNA concentration and purity were analyzed with an Infinite^®^ 200 PRO Nanoquant (Tecan Group Ltd., Männedorf, Switzerland), Qubit 3.0 (Thermo Scientific, Waltham, MA, USA), and a Bioanalyzer Instrument (Agilent Technologies, Santa Clara, CA, USA) with GQN ≥7.

Sequencing was performed on Illumina Miseq (Illumina, San Diego, CA, USA) using Nextera XT v3 600 cycle kit at Fraunhofer Foundation (Santiago, Chile). Samples were amplified by dual-indexing Illumina fusion primers that targeted the 18S rDNA V4 region for eukaryotes [58]. The manufacturer’s recommended protocol was used to perform the sequencing reaction on Illumina Miseq platform. Sequence reads data were archived at NCBI Sequence Read Archive (SRA) with the BioProject number: PRJNA762938.

### 4.7. Metagenomic Analysis

Bioinformatic analysis of Next Sequencing Generation (NGS) data was performed at Genoma Mayor (Santiago, Chile) using DADA2 algorithm pipeline [59]. The DADA2 package uses a parametric model to infer exact amplicon sequence variants (ASVs) from reads, assigning taxonomy to phylogenetically informative marker-gene data, such as the 16S or 18S rDNA gene using the naive Bayesian classifier method [60].

DADA2 was executed as a script in R statistical software v3.5.1 [61]. Reads were quality-filtered and trimmed using the DADA2 function “filterAndTrim”, with options maxEE = 2, truncQ = 2, truncLen = 250 for forward reads and truncLen = 220 for reverse reads. This yielded 4,054,383 quality-filtered non-merged paired-end reads. Error rate models were fitted using the DADA2 function “learnErrors”, separately for each study, and separately for forward and reverse reads. ASVs were then inferred for each sample using the DADA2 functions “dada” with default parameters and paired-end denoised reads were subsequently merged using the DADA2 function “mergePairs” (with options minOverlap = 12, maxMismatch = 0). Chimeric sequences were removed using the DADA2 function “removeBimeraDenovo” (method = “consensus”), separately for each study.

The calculations and drawing graphs were performed using the Phyloseq package version 1.30 [62] available for R statistical software. The microbial diversity indices were studied by calculating the Shannon and Simpson indexes. The diversity comparison among populations was achieved by computing 2D non-metric multidimensional scaling (NMDS) plots based on the Bray—Curtis similarity index. The ASVs were taxonomically identified using SILVA database v.132 as a reference [63].

### 4.8. Statistical Analysis

To identify the main taxa that are changing among samples, two-way analysis of variance (ANOVA) based on Tukey’s multiple comparison tests was carried out at 95% confidence interval [64]. The analysis was carried out using the Real Statistics Resource Pack Software Release 7.8 for Microsoft Excel [65].

## Figures and Tables

**Figure 1 pathogens-11-00494-f001:**
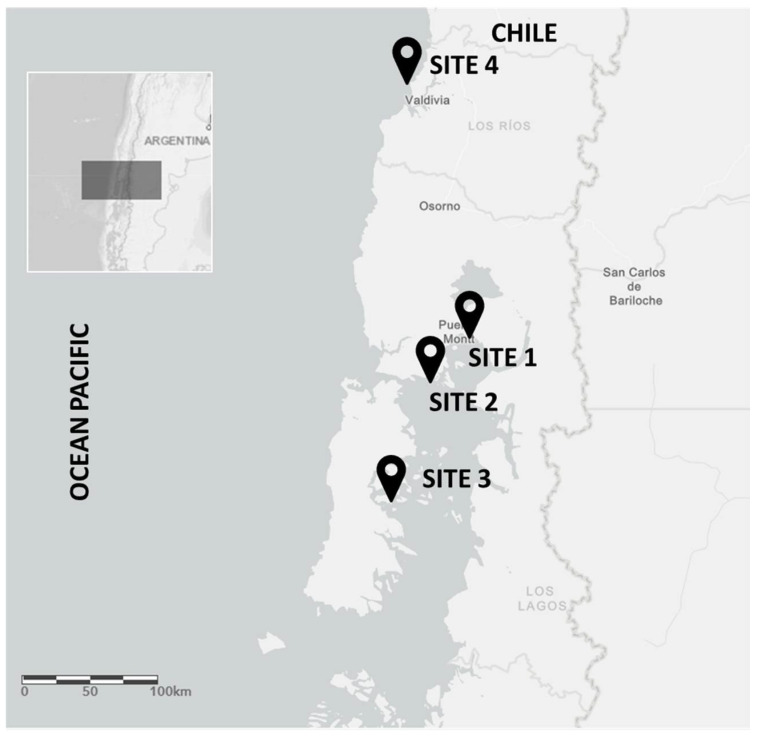
Collection sites for *Mytilus chilensis* samples in southern Chile. Sampling coordinates are indicated in Table 1. Sites 1–3 have mussel farms and Site 4 is free of them.

**Figure 2 pathogens-11-00494-f002:**
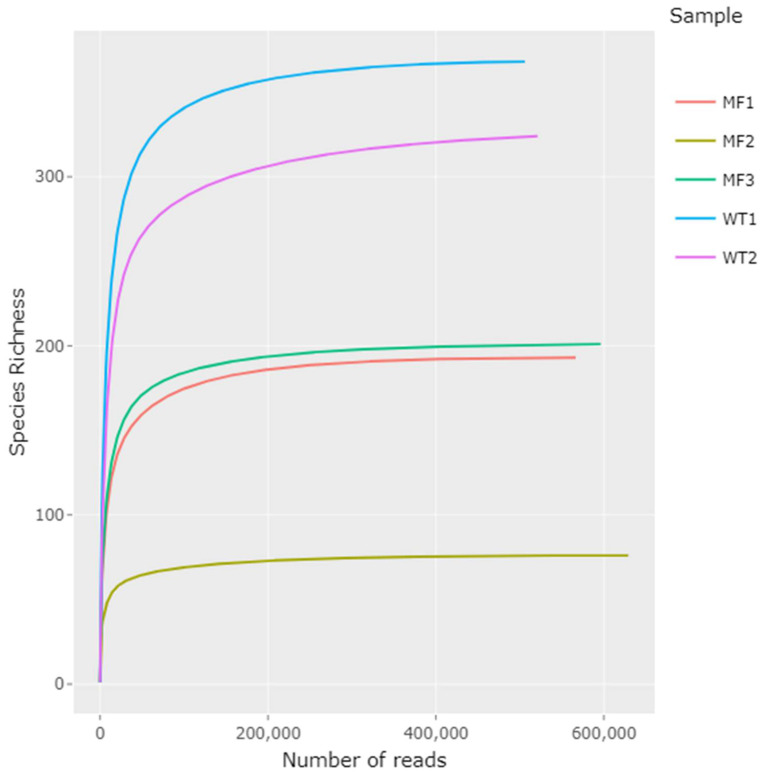
Rarefaction curves showing observed species richness in samples from WT (Wild Type) and MF groups. The MF1–MF3 samples were collected from Site 3 and WT1 and, WT2 from Site 4.

**Figure 3 pathogens-11-00494-f003:**
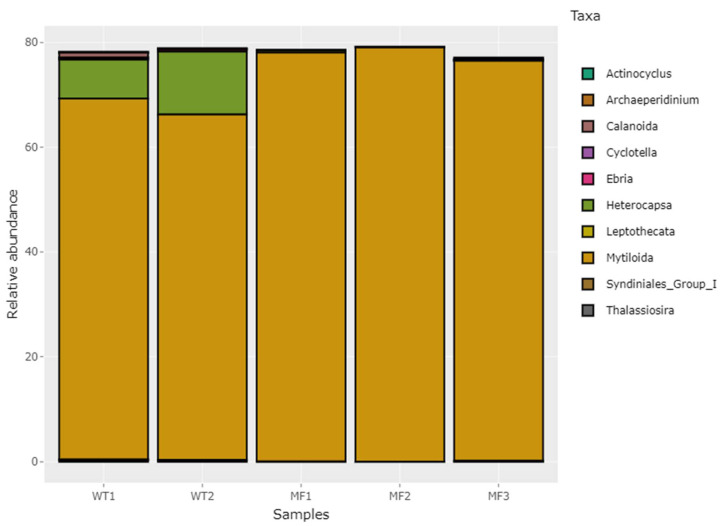
Relative abundances of eukaryotic organisms found in gut of *M. chilensis* by 18S rDNA gene-based profiling analysis. Relative abundance of the top 10 most represented taxa at genus level.

**Figure 4 pathogens-11-00494-f004:**
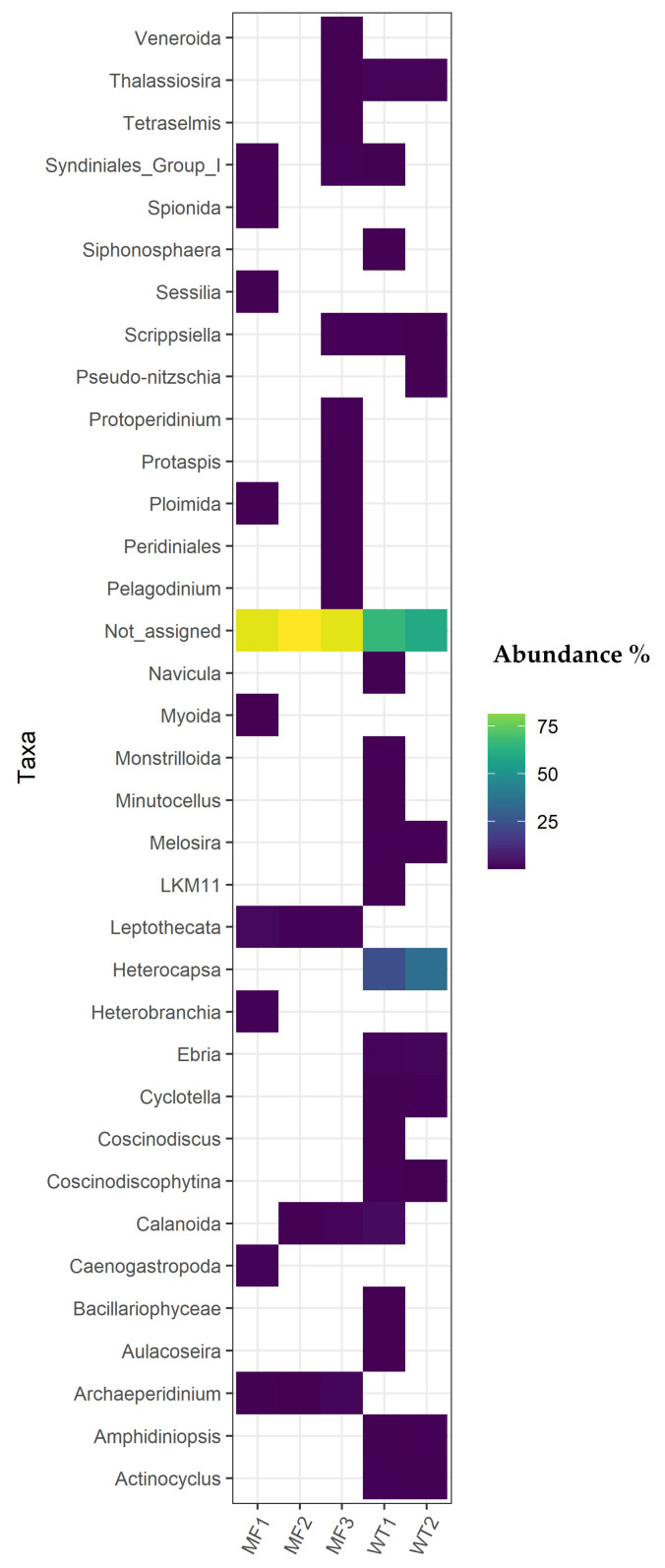
Heat-map (threshold ≥ 0.1%) of eukaryote community at genus level observed in the gut mussel. The change in relative abundance (%) within the community of each phylum is shown by colour intensity. White indicates extremely low abundance and yellow high abundance. The host DNA sequences were removed for visual purposes.

**Figure 5 pathogens-11-00494-f005:**
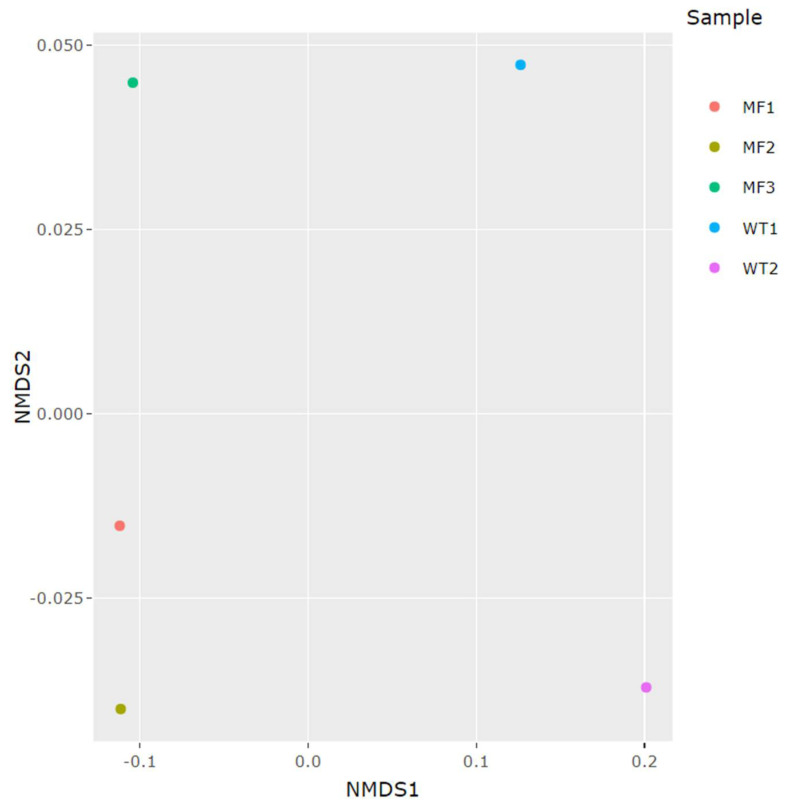
2D NMDS plot of beta diversity for gut samples from *M. chilensis* calculated on Bray-Curtis distance matrix.

**Figure 6 pathogens-11-00494-f006:**
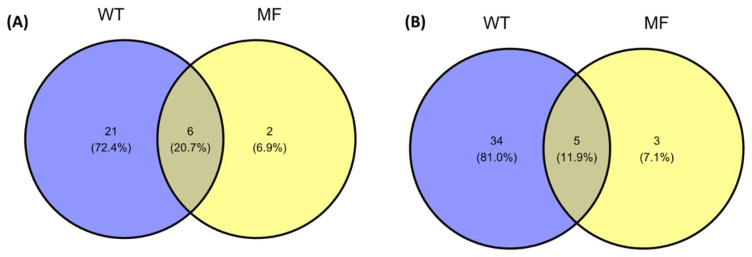
Venn diagram showing the unique, and shared eukaryotic families (**A**) and genera (**B**) among *Mytilus* gut samples.

**Table 1 pathogens-11-00494-t001:** Coordinates and other characteristics of the sampling locations.

Site	Location	Latitude (°S)	Longitude (°W)	Season	PH	Salinity	T °C	O_2_ Dissolved
Site 1	Huelmo Bay	41.677730	73.048031	Spring	8.2	32.5	10.1	8.1
Site 2	Codihue Bay	41.778566	73.373335	Spring	7.7	30.8	9.9	8.4
Site 3	Quinchao Island	42.487768	73.527073	Spring	7.7	31.3	10.2	7.3
Site 4	Calfuco	39.789.288	73.391.704	Spring	7.8	32.5	12.2	7.4

**Table 2 pathogens-11-00494-t002:** Microbiological analysis.

Site	Sample	*E. coli* MPN/100 g	*Salmonella* spp. P/A 25 g	*V. parahaemolyticus* MPN/100 g
Site 1	1	80	Absence	<0.3
	2	50	Absence	<0.3
	3	90	Absence	<0.3
	4	60	Absence	<0.3
	5	80	Absence	<0.3
Site 2	1	80	Absence	<0.3
	2	70	Absence	<0.3
	3	60	Absence	<0.3
	4	80	Absence	<0.3
	5	70	Absence	<0.3
Site 3	1	20	Absence	<0.3
	2	10	Absence	<0.3
	3	<0.3	Absence	<0.3
	4	10	Absence	<0.3
	5	<0.3	Absence	<0.3

MPN: most probable number. P/A: Presence/Absence.

**Table 3 pathogens-11-00494-t003:** Pathogens DNA detection results.

Site	Pathogen	Samples Analyzed	PCR Detection *
S1	*Bonamia* spp.	30	n.d.
	*M. refringens*		n.d.
	AbHV		n.d.
	*X. californiensis*		n.d.
	OsHV-1		n.d.
	*Perkinsus* spp.		n.d.
S2	*Bonamia* spp.	30	n.d.
	*M. refringens*		n.d.
	AbHV		n.d.
	*X. californiensis*		n.d.
	OsHV-1		n.d.
	*Perkinsus* spp.		n.d.
S3	*Bonamia* spp.	30	n.d.
	*M. refringens*		n.d.
	AbHV		n.d.
	*X. californiensis*		n.d.
	OsHV-1		n.d.
	*Perkinsus* spp.		n.d.

*n.d: No detected.

**Table 4 pathogens-11-00494-t004:** Characteristics of 18S rDNA metagenomic libraries.

Site	Sample *	Input	Reads Filtered	Reads Merged	Non Chimeras	No. of Genera	No. of Family	Shannon-Wiener Index	Simpson Index
3	MF1	1,038,053	828,653	737,414	566,455	27	26	3.11	0.95
3	MF2	942,997	789,981	788,394	629,065	14	11	3.00	0.94
3	MF3	994,332	820,426	817,042	596,117	31	27	3.18	0.95
4	WT1	999,049	788,556	782,046	506,051	63	49	3.46	0.96
4	WT2	1,042,958	826,767	821,658	521,142	66	42	3.41	0.96

* WT = Wild Type, MF = Mussel Farm.

## Data Availability

Raw sequences have been submitted to the NCBI sequence read archive (SRA) database under the BioProject number: PRJNA762938.

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
