# Peer review of "Health Status of Mytilus chilensis from Intensive Culture Areas in Chile Assessed by Molecular, Microbiological, and Histological Analyses"

_pathogens, 2022, doi:10.3390/pathogens11050494_

Round 1

Reviewer 1 Report

The manuscript entitled “Health status of Mytilus chilensis from intensive culture areas in Chile assessed by molecular, microbiological, and histological analyses.”investigates the health status of Mytilus chilensis farmed in Chile . The manuscript is generally well written and provides new information on the circulation of various pathogens in Mytilus chilensis in an important farming area in Chile. However there are some comments and mistakes that should be addressed before pubblication.

Overall.

In this work the sampling is punctual in the only month of October, why did you choose this month? It would be more interesting to carry out sampling throughout the year because conditions can change and cause pathogens to emerge.

Regardinig B. exitiosa for example OIE (in Manual of Diagnostic test for Aquatic Animals) reported....... …...Prevalence is variable in O. chilensis (from 0% to nearly 80%) (Cranfield et al., 2005; Diggles et al., 2002). In the Southern Hemisphere, infection with B. exitiosa shows the highest prevalence from January to April, with the parasite barely detectable in September and October (Hine, 1991a). Stressors such as exposure to extreme temperatures (below 7 C or above 26°C) and salinity (40%), starvation (prolonged holding in filtered sea water), handling (vigorous stirring four times per day), or heavy infection with an apicomplexan (Hine, 2002), can affect the disease dynamics of B. exitiosa in O. chilensis (Hine et al., 2002).

Regarding Marteilia refringens OIE (in Manual of Diagnostic test for Aquatic Animals) reported …..The threshold temperature for parasite sporulation and transmission is 17°C. T …...

Therefore also OsHv-1 infection is influenced by seawater temperatures....for example in France mass mortalities of young Crassostrea gigas have notably affected all rearing sites along the coasts where seawater temperatures reach 16 to 17°C (Bedier 2010, EFSA 2011). The high seawater temperatures of spring and summer months could promote the production of viral particles in association with massive mortality among Pacific oyster spat (Garcia et al. 2011). Moreover, from a study conducted in laboratory, it is emerged that temperature is one of the major factors acting on the mechanisms of OsHV-1 infection (Sauvage et al. 2009).

In this study in October the seawater temperature was 9.9 ° C -12 ° C, it would be interesting to monitor the health of the shellfish throughout the year to see how the effect of higher and lower temperatures could influence and stress the bivalves with consequent weakening of the immune system of these animals and favor the onset of diseases

Title

Mytilus chilensis should be written in italics.

Abstract

Line 26, 27….“spp.” should not be written in italics. Check it throughout the text.

Line 28 and 33….change “sanitary” with “health”.

Introduction

Line 60….“spp.” should not be written in italics.

Materials and Methods

According to the guidelines of the journal, the order of Research Manuscript Sections is:

Introduction, Results, Discussion, Materials and Methods and Conclusions (This section is not mandatory, but can be added to the manuscript if the discussion is unusually long or complex.).

So Materials and methods must be moved to the bottom of the manuscript and consequently double check the number to be attributed to figures and tables.

Line 94…. here it is said that oxygen concentration and oxygen saturation levels were monitored but in table 2 is reported only the data for dissolved oxygen. Please correct it.

Line 103 and 107…. the year is not reported in the ISO methods mentioned.

Line 106, 108, 163, 184….“spp.” should not be written in italics. Check it throughout the text.

Line 150… Previous results… if a publication has been made on these data, put a bibliographic reference.

Line 193. Remove a space between number and S.

Line 127, 136…..Sometimes only OsHV appears in the text, sometimes OsHV-1 please uniform it throughout the text. I think it is more correct to put OsHV-1.

Results

The section from line 254 to 259 in my opinion should be moved to materials and methods, and maybe even from line 253 to 254.

Line 260….turbidity….this parameters is not reported in table 2

Line 273 ….“spp.” should not be written in italics

Line 283….check OsHV and spp.

Line 300…… the abbreviations MF and WT appear here for the first time, move here the explanation present in line 342 and 343

Line 301, 302,303…… in number change the point in place of the comma

Line 329, 330,333…………In percentage values change the point in place of the comma

Lines 276-277 and table 3

Category A of mussel farms must have a maximum of 230 MPN of E. coli for every 100 grams of mollusk flesh and intervalvular liquid, considering 5 tubes and 3 dilutions (sernapesca, 2018), so in table 3 results for E.coli are correct? If you find 80 MPN/g, in 100 gr E. coli was 8000 beyond the limit for category A.

In line 276 is correct <100 CFU/g? or it must be <100MPN/100g….please correct or explain better.

Table 1:

Correct OsHV-u1.

Perhaps it would be better to include in the table also the bibliographic references for the disease information

Table 2

O2 change with O2

Table 3

Scientific name should be written in italics.

Site 4 results are missing.

MPN/g is correct?

Table 4

Site 4 results are missing.

Correct OsHV-u-1.

Table 5, Figure 3 and 4

Insert the explanation of the abbreviations MF and WT in the caption

Discussion

Lines 378, 480, 494, 500….change “sanitary” with “health”.

From line 472 to 479….the font is bigger.

Line 482 ….“spp.” should not be written in italics

Line 485 correct <100 MPN/g with <100 MPN/100 g

References

References number 26 and 55 are missing in the text.

Author Response

Dear Reviewer,

Here are the answers: 

Q1: In this work the sampling is punctual in the only month of October, why did you choose this month?: A: This month was selected because is when de mussels processing plants start their activities, being the biomass higher than other months. In addition, this study was a firts approach to asses the health status of M. chilensis mussel, the time variable is interesting to be considered in future evaluations. 

The others suggestions were considered in the manuscript. 

Reviewer 2 Report

The manuscript titled "Health status of Mytilus chilensis from intensive culture areas in Chile assessed by molecular, microbiological, and histological analyses" aims to evaluate the health status of M. chilensis and to study its potential as a host species for exotic diseases by microbiological, molecular, and  histological analyses.  The introduction provided sufficient background and include relevant and current references. The methodology is appropriate and complete using a diversity of techniques to detect and identify pathogens. However, this kind of study requieres at least one year sampling, in order to detect parasites or pathogens that could be influenced by high temperatures or other environmental fluctuations. Here, one puntual sampling during spring was done. The results and the discussion shoud be given in the order that the methodologies were performed. In my opinion, a histolopathological image of the procaryotic inclusions should be given. In the reference list all the specific names should be written by italic.

Line 60 and from here on: re write "spp" without italic

Line 84: replace the word "around" by "approximately"

Line 286: about low and the moderate intensity: there are not results about these values, and in material and methods the intensity is not calculated.

Line 426: Abbreviate Mytilus: M.

Line 439: ...epithelium that affects: I suggest to writing before affects: "could affect"

Line 470: Abbreviate Mytilus: M.

Line 453: The cite: Lieverloo et al 2012: is not in the reference list.

Line 472-479: the size of the words is different

Line 495: Abbreviate Mytilus: M.

Line 496: delete once: on the list

References:

All the specific names should be written by italic.

Line 518: the name of the journal should be written by italic and there should be a comma after the volume.

Line 521: the name of the journal should be abbreviated.

Line 542: this reference should be written after the reference of the line 544. The name of the journal should be abbreviated.

Line 575: Gracia Villalobos is not cited in the text.

Line 581: The year of the publication should be written after the names of the authors.

Line 604: this cite should be written after the cite of line 607.

Line 618: Rewrite the names of the authors.

Line 622: The name of the journal should be written by italic and abbreviated.

Line 55: The cite Ottaviani E (2011) is not in the text.

Line 66: The cite van Lieverloo et al (2012) is not in the text.

Line 677: The cite Vázquez N, Cremonte F (2017) is not in the text.

Author Response

All suggestions are accepcted and modified in the manuscript.

Best regards

Reviewer 3 Report

Letter to Authors
pathogens-1493372-v1
Health status of Mytilus chilensis from intensive culture areas in Chile assessed by molecular, microbiological, and histological analyses.
P. Santibanez, J.L. Romalde, A. Figueras, D. Fuentes, J. Figueroa

211029

Dear authors,
You have successfully shown the sanitary and infectious status of Mytilus cultured in Chile. Your methodology and results are applicable to shell culture practices in some other temperate seas. It is thus worth publishing your research in an international journal like Pathogens, if adequately prepared. Your current MS needs, however, substantial revision. Several redundant statements exist in the introduction and discussion sections. Method description is incomplete regarding particularly on your sample pooling scheme. Some parts of the results section are redundant with the method section. You seem unable to write Latin names and taxon ranks correctly. Use of abbreviations is confusing. The current MS has many typos and syntax errors, and thus extensive English editing is necessary. I am sorry to say I suggested "major" this time. See below for detail. Words in braces indicate options.

L2
Mytilus chilensis -> in Italics

L26,27,etc
spp (English) -> in Roman
Check thoroughly.

L35 keywords
Mytilus chilensis -> replace
Do not list words which appear also in the title. Duplicate hits upon computer search do not make sense. Give words that do not appear in the title to draw attention from wider readership. Posting words that neither appear in the abstract is better, because even in full-text search/indexing robots may not weigh much on words deeper (posterior) in the text. Hint: bivalve, Rickettsia-like, protozoa, DNA barcoding, sanitary status, etc.

L78
The Table 1 shown .. pathogens investigated. -> delete
Wording like "X is shown in figure/table Y" imposes killing readers' times to read such an information deficient sentence telling only that there is a figure/table. You should present an outline or a perspective drawn from the figure/table and cite it in parentheses at the end.

L92
The Table 2 shown .. for each mussel farm. -> delete
You may cite table 1 together with figure 1 at the end of 2nd sentence of this paragraph. You may cite this table again on L95.

L98
processing -> processed

L100
a composite of individuals ?
Your pooling scheme is unclear. How many individuals/tissues were pooled?

L103
"Microbiology of .. Escherichia coli" (verbose) -> delete
It is better present an internet URL.

L106
URL ?

L107
"Microbiology of .. Salmonella spp" -> delete
URL ?

L110
bacterial -> bacteria

L118
DNA extractions -> extracted DNA

L121
summary -> summarized

L133
see -> delete

L136,414
OsHV virus -> OsHV
Not "Ostreid herpesvirus virus".

L140,155,168,187,189
(Agilent, USA) -> delete

L149
Bacteria -> Bacterial

L157,179
are -> were

L161
agarose gel 2% -> {2% agarose gel, agarose gel of 2%}
See L147.

L160,182,191
(Biotium, Hayward, CA) -> delete

L165
object -> objects
study in (wordy) -> delete

L168
contains -> contained

L170
consist -> consisted

L176,186
contain -> contained

L178
consists -> consisted

L183,192
a -> an

L193
18. S rDNA -> 18S rDNA

L194
pooled samples ?
The pooling scheme is unclear. You extracted DNAs form gut of the shell independent from specific parasite detection (L110).

L201
Eukaryotes (English) -> lower case

L210
DADA2 -> DADA2 v1.14
(Divisive Amplicon Denoising Algorithm 2) (redundant) -> delete

L215
DADA2 v1.14 -> DADA2

L228
population comparison -> diversity comparison among populations

L229
non-metric MDS -> non-metric multidimensional scaling (NMDS)
Spell-out at the first place. See L360.

L240
knife -> a knife

L250-254
In this study, .. in Table 1.
Move to the method section 2.2 with Table 1. Some revision may be necessary to fit with this re-organization. Revision of the current L253 is needed because wording like "X is shown in figure/table Y" imposes killing readers' times to read such an information deficient sentence telling only that there is a figure/table. You should present an outline or a perspective drawn from the figure/table and cite it in parentheses at the end.

L254-259
The site S1 is .. in the area.
Move to the method section 2.1.

L263 table 1 body
Insert a blank line between description of each species on the "Disease information" column to make it clear about the beginning and end of each description.

L269
For the analyses .. summarized in Table 3. -> delete
You may cite the table at L274.

L278 table 3 heading
E. coli, Salmonella, V. parahaemolyticus -> in Italics
MPN -> CFU/g ?

L280
Histological -> in lower case
Make sure whether sub-section titles are in title case or lower case. This title is neither.

288
not observing structural alterations -> observing no structural alteration

L289
showed -> shown

L300
MF -> mussel farm (MF)
WT -> control (CT)
Spell-out at the first place. Use of "WT group" in your current MS means "control group group". Use CT instead of WT throughout including figures and tables. If you do not like to replace WT throughout, you may describe it as "wild type (WT)" here.

L304
show -> showed

L305
are -> were

L306
remain -> remained

L307-312
Differences in .. packages in R. (redundant) -> delete

L315
Tukey HSD -> Tukey's
Spell-out of HSD is necessary but verbose.

L321
Make sure whether sub-section titles should be in title case or lower case. Preceding titles in M&M section are all in lower case.

L334 Fig.3 picture
Two sub-figures seem identical. Either of the two can be omitted, and the legend should be revised accordingly.
Fonts in the figure picture are too small to see. They must al least be equal to or larger than the fonts used in the main text.

L339
The heat-map (Figure 4) shows .. relative abundance. -> delete
You may cite Figure 4 at the end of next sentence: "as shown in a heat-map (Figure 4)".

L343
the control group (WT) and the mussel farm (MF) -> CT and MF groups
See L300.

L344
Peridiniales -> in Roman
Family names should be in Roman.

L347
was previously .. Liu et al., 2020) (redundant) -> delete
See L468.

L360
Beta diversity .. in the data. (redundant) -> delete

L362
Eukaryotic (English) -> in lower case

L367
the control -> CT
the mussel farm -> MF
Use abbreviations consistently from 2nd appearance and later.

L370 Figure 5 picture
Change font sizes al least to equal to or larger than those used in the main text.

L372
distance -> similarity
See L230.
(WT = Control mussel, MF = Mussel farm) (redundant) -> delete

L377
mussels farms -> mussel farms
See also L434.
shows -> showed
has -> had

L379
findings -> symptoms ?
and  -> nor

L382
do not reflect -> have not reflected

L384
mussels, -> mussels. (break sentence here)

L385
M. -> Marteilia
Avoid complication with Mytilus throughout. Or otherwise, you may abbreviate it as Ma. while Mytilus being My. throughout.
Paramyxea -> in Roman

L399
2009), -> 2009). (break sentence here)
possible .. possible (repeated word) -> possible .. putative ?

L406
control -> CT

L407
parasite -> parasitize

L408
Oyster Herpesvirus -> OsHV
See L127.
Abalone herpesvirus-1 (AbHV) -> AbHV
See also L419.

L411
The Oyster Herpesvirus type 1 (OsHV-1) -> OsHV-1

L413
Pacific oyster (English) -> in Roman

L419
Abalone herpesvirus-1 (AbHV) -> AbHV

L422
Malacoherpesviridae -> in Roman
Family (English) -> in lower case

L423
presence of (wordy) -> delete

L426,470,495
Mytilus chilensis -> M. chilensis

L431
X. californiensis .. in this study (does not make sense) -> presence of X. californiensis was surveyed in this study

L432
not detecting presence of DNA of this bacterium (not a complete sentence) -> but DNA of this bacterium was absent

L434
founded -> found

L435
2022), -> 2022).
more (childish) -> further

L436
the genus (wordy) -> delete

L443-445
including .. larger colonies (redundant) -> delete
Do not make a simple reference list (A stated this, B analyzed that, C argued it, or alike). Use noun phrases to make abstract contents of those references.
I think other parts in this section also contains simple reference lists that undersells your own research.

L448
The authors -> We
Readers will misunderstand who are the "authors".

L450-460
The eukaryotes include .. distribution (Manichanh et al., 2008). (redundant) -> make it short <100 words or delete
These statements are suitable for an introduction section.

L464
Family -> in lower case
Peridiniales -> in Roman

L467
Heterocapsa genus -> {genus Heterocapsa, Heterocapsa}

L472-479
Use a smaller font as other part of the text.

L488-492
(Oliva, et al., 1986) .. (Cremonte et al., 2015) -> (Oliva, et al., 1986; Mix and Breese , 1980; Campalans, et al., 1997; Cremonte et al., 2015)
Do not make a simple reference list.

L515 references
Check the reference list carefully again from the beginning. Reference lists are frequently hotbeds of errors. You might add, omit or swap citation in the main text on the way internal revision. Some inconsistency might occur where reference(s) is absent form the list or excess listed. If so, readers think you are making irrelevant citation. It is the authors' responsibility that all references are properly cited.
I think you do not follow the journal style both in the main text and the list. 

L542
i ?

L665,667
Sernapesca -> in upper case
See L77.

and more ..

Author Response

(The authors gave the same response as above.)

Round 2

Reviewer 1 Report

Dear Authors, the revised article “Health status of Mytilus chilensis from intensive culture areas in Chile assessed by molecular, microbiological, and histological analyses” is more understandable and clearer than the previous version. However, there are still some issues in the manuscript that need to be modified.

Introduction

Line 45….“mollusks species” please correct it.

Materials and Methods

According to the guidelines of the journal, Materials and Methods is the Section number 4 after discussion.

So Materials and methods must be moved to the bottom of the manuscript and consequently double check the number to be attributed to figures and tables.

Line 80 to 98…… this section can be improved; it is redundant and not clear. Moreover, explain well that the search for bacteria, viruses and protozoa was performed only in sites 1,2 and 3.

Line 135 correct “Ostreid (OsHV-1) herpesvirus” with….. ostreid herpesvirus 1 (OsHV-1)

Line 146 there is a parenthesis after [19], remove it.

Reference [20] is missing in the text…..check it.

Line 197. Remove a space between number and S.

Line 213. NGS appears for the first time, please explain it in extended and then abbreviation

Results

Line 257 line….. bracket, move above

Table 2

correct E. coli MPN/g with E. coli MPN/100 g

Line 288-289…… the abbreviations MF and WT appear here for the first time, move here the explanation present in line 304 and 305

Line 320, 321, 323,324…………In percentage values change the point in place of the comma

Figure 3

Legends are too small, difficult to read

Discussion

Line 366 bracket, remove it.

Line 367 insert some references after …..publications and studies.

Line 395 ….“spp.” should not be written in italics

Line 412 insert a space between…. study.DNA

From line 445 to 452….the font is bigger.

References

 Line 498 ….“spp.” should not be written in italics

Line 528 Salmo salar should be written in italics

Reference number 27 it is correct? Check it please

Author Response

Dear Reviewer

I am attaching  a new version of the maniscript. 

Main modifications

The materials and methods were moved according  to the guidelines of the journal. The grammar and writing was improved according to the observations of all reviewers. 

Best regards

Reviewer 3 Report

Letter to Authors
pathogens-1493372-v2
Health status of Mytilus chilensis from intensive culture areas in Chile assessed by molecular, microbiological, and histological analyses.
P. Santibanez, J.L. Romalde, A. Figueras, D. Fuentes, J. Figueroa

211211

Dear authors,
I am sorry to read your v2 MS ignorant of my comments of importance last time. The current delivery is hardly improved from the last time. Your sample pooling scheme is still unclear. This incomplete method description means "control missing" and thus one more round of major revision is necessary. Many other miscellaneous issues of grammatically questionable and nonsensical wording still remained or newly introduced. Be careful writing logically. I do not know if you consulted an English proofreader, but I suggest you to do it after you finished revision. See below for detail.
Words in braces indicate options. Those in brackets can be omitted.

L45
mollusks species -> {mollusks, mollusk species}

L47
in the Bivalvia Class -> {in Bivalvia (Latin), in the bivalve class (English)}

L81,134,262
mussel farms -> mussel farms (MF)

L82
control mussel group -> control mussel group (wild type, WT)
You are refractory for replacing WT with CT throughout introducing new inconsistency.
mussels farm -> MF

L84
with a high anthropogenic activity (redundant) -> delete

L89
mussel farms -> MFs

L90
control population -> a WT population

L91
Mussels were collected .. -> new paragraph
transferred alive .. for transit to the laboratory (redundant) -> delete

L93
transported -> kept cool and transported
processed -> processing

L105
mussel farming -> MF

L106
OIE -> OIE [19] ?
SERNAPESCA -> SERNAPESCA [12]

L107
using as reference the National Fisheries and Aquaculture Service of Chile -> following SERNAPESCA ?

L109
according to the microbiological analysis (unclear and redundant) -> delete

L110
beta-glucuronidase - positive determination (does not make sense and verbose) -> delete

L118
parasites -> eukaryotic parasites

L119
extracted -> {sampled, isolated, taken}

L121
of detect DNA target -> {to detect DNA target, for detecting DNA target, of DNA target detection}

L124
the extracted DNA -> {extraction of DNA, DNA extraction}

L126
Each sample of DNA extractions consists of a composite of
Your sample pooling scheme is unclear. What is "sample of DNA extractions (nonsensical?)"? Was DNA extraction done individually or on pooled tissue samples from three individuals?
consists -> consisted (you may ignore this if revision over here were thoroughly omitting this word)

L127
PCR reactions with primers and conditions summarized in Tables S1,S2
duplicate -> duplicate. (break sentence here)

L128
The primers .. respectively. (redundant) -> delete
Wording like "X is shown in figure/table Y" imposes killing readers' times to read such an information deficient sentence telling only that there is a figure/table.

L130
Me15/Me16 primers .. PCR inhibitors. (verbose) -> Me15/Me16 primers [18] worked for an internal control.

L137-138
ORF77F1/ORF77R1 and ORF77 FAM-probe -> list the sequences in Table S1 (optional)

L140
is -> was

L141
have been -> being
in the appropriate concentrations (see Table S2) (redundant) -> delete

L146
[19]) -> [19]

L160
Test[19] -> Test [19]

L161
R1 ?
This is absent from table S1.

L166
An aliquot .. gel electrophoresis (redundant) -> An aliquot of each PCR reaction product was checked by [[2%] agarose] gel electrophoresis

L172
Perk ITS ? (PerKITS in table S1)

L173
the Manual of 173 Diagnostic Test -> {a manual, the Manual of 173 Diagnostic Test for Aquatic Animals 2021}
I suggest the former because of compactness.

L178
Following amplification, .. agarose gel (redundant) -> The PCR products were [subsequently] checked by [[2%] agarose] gel electrophoresis

L181
Manual of Diagnostic Tests for Aquatic Animals -> {manual, Manual of Diagnostic Tests for Aquatic Animals 2021} [19] ?

L187
The PCR products .. a UV illuminator (redundant) -> The PCR products were [subsequently] checked by [[2%] agarose] gel electrophoresis

L193
(Agilent, USA) -> delete
See L143.

L195
After amplification .. a UV illuminator (redundant) -> The PCR products were [subsequently] checked by [[2%] agarose] gel electrophoresis

L198
Make your pooling scheme clear. See sub-section 2.3.

L205
Eukaryotes (English) -> in lower case

L205-208
E572F: 5'- .. -3' -> move to Table S1 presenting a complete primer set for pathogen detection of three categories (virus, bacteria and eukaryotes)

L213
NGS -> next-generation sequencing

L217
RDP -> delete

L228
graphs -> drawing graphs

L235
in the samples -> among samples
You did not make timeline sampling.

L239-251
Move this sub-section to L103 to fit with the order in results.

L254-256
Physico-chemical (Table 1
) (then what?) -> revise
Mention "then what" matters. Wording like "X is shown in figure/table Y" imposes killing readers' times to read such an information deficient sentence telling only that there is a figure/table. You should present an outline or a perspective drawn from the figure/table (then-what matters).

L258-261
For the analyses .. public health. -> delete or move to the method section with revision. Reference #33 may be cited in the method section.

L261
The results are consistent with -> Analysis of foodborne pathogens in this study corroborated the

L273
analyzes -> analysis

L274
is -> was
Results should be in past tense, because exact reproducibility is not guaranteed. It was that at that time, but now it may not be exactly the same at present.

L276
do -> did

L277
alteration (repeated word) -> anomaly ?

L278
show -> shown
are -> were

L282
copurify -> are copurified

L296-301
To assess .. Phyloseq in R. -> move to M&M section or delete.

L304
Wild Type group, WT -> WT group

L305
farm -> farms

L312
control -> WT
group -> groups

L313
M. chilensis -> in Roman
Roman words in fully Italicized line are equivalent to Italics in a Roman line. This is the formal scripting.

L318-319
The figure 3 shows .. in the mussels gut. (then what?) -> delete
You may cite figure 3 at the end of next sentence.

L321
79,02% -> 79.02%
So use a dot instead of a comma as in L322,323.

L325
Fig.3 picture
Two sub-figures seem identical. Either of the two can be omitted, and the legend should be revised accordingly.
Fonts in the figure picture are too small to see. They must al least be equal to or larger than the fonts used in the main text.
You seem dishonest telling "all suggestions are accected and modified".

L329,346,357
(WT = Wild type, MF = Mussel farm). (redundant) -> delete
See figure 2 legend.

L334
of -> {in, from}

L335
individuals from mussel farms -> MF group

L362
mussel farms ?
Make sure if sampling sites were exactly mussel farms. If so, you should revise sub-sections 2.1 and 3.2, figure 1, and tables 1-3. Further revision throughout may be necessary to fit with each other. Do not introduce excess in-house terms that disturb readers' short term memory when reading.

L365
genus -> genera
are -> were

L367
What are "the publications and studies"? References are needed.

L369
mussels, -> mussels. (break sentence)
OIE -> OIE [19] ?

L364-383
First, .. is relevant. -> revise or delete
Do not make a simple reference list (A stated this, B analyzed that, C argued it, or alike). Use noun phrases to make abstract contents of those references.
The simplest way, I think, is to cite references #21,23,37-46 at the end of previous sentence in parentheses deleting this long tedious simple reference list.

L386
metabarcoding, results in accordance with (incomplete sentence) -> metabarcoding in accord with

L387
What is "bibliographic status"?

L388
Family -> family
Perkinsidae -> in Roman

L389
parasitize genus of (verbose) -> delete
alveolates -> alveolate
parasite (syntax error) -> parasitizes
Reference is needed.

L390
Second, at the virus level, -> delete
The rest of this sentence needs reference(s).

L391-395
Due to the proximity .. Pecten maximus. -> delete
This tedious long story is worse than a simple reference list. Absence of reference indicates this story is of your own research or of plagiarism.

L395
spp. -> in Roman

L396-403
The Mytilus species are .. with other viruses [51]. -> delete
References #48-50 may be cited at the end of previous sentence.

L406
supporting our findings -> in accord with our results.
Emptiness cannot support something.

L407
(WS) -> delete

L408
organism (WS-RLO). The causative agent, -> delete

L409-412
occurs along .. in this study. -> delete
This tedious long story is worse than a simple reference list.

L413
previous studies -> a previous study
You cite only one study [11] here.

L414
studies -> study

L416-428
Move this paragraph to the top of this section to fit with the order in the result section. Some revision may be necessary to fit with this re-organization.

L418
could affects -> affecting

L419
mortalities were (syntax error) -> mortality was

L420-421
The occurrence .. is widespread among mollusks.
Reference(s) needed.

L422-424
including .. gland tubules -> delete
Do not make a simple reference list.

L438
the control -> WT

L446
are -> delete

L451
which is reflected in -> which reflected

L486 references
Check the reference list carefully again from the beginning. Reference lists are frequently hotbeds of errors. You might add, omit or swap citation in the main text on the way internal revision. Numbering of the references might then shift. If so, readers think you are making irrelevant citation. It is the authors' responsibility that all references are properly cited.

L514
author? title?

L533
Aquatic Manual Online Access - OIE - World Organisation for Animal Health -> OIE - World Organisation for Animal Health. Manual of Diagnostic Tests for Aquatic Animals 2021 (author. title)

L550
Nai¨veNai¨ve -> Naive
Make sure if paper titles are in lower case. Check thoroughly.

L551
+ Downloaded from ?

L554
Journal title?

Author Response

Dear Reviewer

I appreciate your suggestions, comments, and observations to improve the manuscript.

Main modifications: The materials and methods were moved according to the guidelines of the journal. The grammar and writing was improved according to the observations of all reviewers. A major restructuring of the manuscript was carried out focused on your observations.

L110 beta-glucuronidase - positive determination (does not make sense and verbose) -> delete. Response: It is the exact ISO standar title 

Best regards

Round 3

Reviewer 3 Report

Letter to Authors
pathogens-1493372-v3
Health status of Mytilus chilensis from intensive culture areas in Chile assessed by molecular, microbiological, and histological analyses.
P. Santibanez, J.L. Romalde, A. Figueras, D. Fuentes, J. Figueroa

211227

Dear authors, your r3 MS still needs minor revision. You have re-organized putting M&M away. You should revise spell-out and abbreviation of terms according to your re-organization in which M&M is put away. Some other miscellaneous points also remained or newly introduced. See below for detail.

L46
(Hupe 1854)
Make sure if parentheses are necessary. Parenthesizing author names has a particular meaning in zoological nomenclature.

L47
Bivalvia -> in Roman
You seem unable to script Latin names properly.
Class (English) -> lower case
You seem unable to distinguish Latin and English names.

L52,98
mussel farms -> mussel farms (MFs)

L60,80,83,86,243,262
mussel farms -> MFs

L86
category A (punctuation) -> category A.

L105
AbHV and OsHV-1 -> abalone (AbHV) and Ostreid herpevirus 1 (OsHV-1)

L117-120
Mollusks secrete .. PCR applications [13,14]. -> move to L124 deleting the 1st sentence
Do not make a long unsightly excuse. Reference #12 can be cited at the 2nd (new 1st). Put essential matters (your results) first.

L142
Wild Type group (WT) and Mussel Farm (MF) -> WT (Wild Type) and MF groups

L222
has -> have
"spp." is plural (several species).
genus Mytilus spp. -> genus Mytilus.

L227
studied -> studied,

L232
Mytilus spp. genus -> genus Mytilus
being consider -> being considered

L259
specie Mytilus chilensis -> species M. chilensis

L351
abalone (AbHV) and Ostreid herpevirus 1 (OsHV-1) -> AbHV and OsHV-1

L415
(Wild Type) -> delete

L432
nai:ve ->naive

Author Response

Dear Reviewer

We appreciate your important comments on the manuscript. We have two observations that were not processed. 

L52,98: mussel farms -> mussel farms (MFs)

L60,80,83,86,243,262 mussel farms -> MFs

Reason: The manuscript was completely revised. The abbreviation MF is exclusive for samples from site 3. The text has been corrected because it confuses the reader by assuming that MF corresponds to the sampling sites rather than samples from a specific location (site 3).

Best regards
